# A Robust Multivariate Time Series Classification Approach Based on Topological Data Analysis for Channel Fault Tolerance

**DOI:** 10.3390/s25092709

**Published:** 2025-04-24

**Authors:** Seong-Yeon Jeung, Jang-Woo Kwon

**Affiliations:** 1Department of Electrical and Computer Engineering, Inha University, Incheon 22212, Republic of Korea; ran22314@inha.edu; 2Department of Computer Engineering, Inha University, Incheon 22212, Republic of Korea

**Keywords:** rotary machine, deep learning, TDA, channel fault, MTSC

## Abstract

In this study, we propose a robust artificial intelligence (AI) model for vibration monitoring of rotating equipment to support reliable operation across various industries, including manufacturing, power plants, and aerospace. The reliability and completeness of sensor data are essential for early detection of anomalies in equipment and for performing predictive maintenance. While AI-based predictive maintenance and condition-monitoring technologies have advanced in recent years, the issue of data loss caused by sensor failures remains a significant challenge that leads to performance degradation of AI models. In particular, for equipment utilizing multiple sensors, the complete loss of data from a single sensor significantly diminishes the predictive maintenance capability of AI models, thereby reducing their reliability. To address this issue, this study introduces topological data analysis (TDA) to develop a robust AI model. TDA analyzes the topological structure of sensor data to generate consistent feature vectors that capture the intrinsic characteristics of the data. This enables stable predictions even when certain channels of multi-sensor data are entirely missing. The proposed method demonstrates high performance resilience under conditions of partial sensor data loss, thereby contributing to enhanced reliability of AI-based predictive maintenance systems and the establishment of efficient maintenance strategies in the future.

## 1. Introduction

Rotating equipment plays a critical role across various industries, including manufacturing, power plants, and aerospace. These systems are susceptible to a range of mechanical and electrical anomalies, which can lead to performance degradation, emergency shutdowns, or, in severe cases, catastrophic failures. As such, continuously monitoring equipment, detecting anomalies early, and performing predictive maintenance are essential tasks for ensuring stable operation. This necessity has driven advancements in the field of Prognostics and Health Management (PHM).

In recent years, research in PHM has increasingly leveraged artificial intelligence (AI) algorithms to detect anomalies in equipment based on sensor data [1,2]. The goals of these studies are to detect performance degradation or faults in advance and maximize equipment uptime through predictive maintenance. Commonly, sensors such as vibration, temperature, and pressure sensors are used to monitor equipment conditions in real time. The collected data are analyzed to detect anomalies and predict the optimal timing for maintenance [3,4]. Therefore, accurate sensor data are critical for ensuring the stability and efficiency of equipment operation.

However, the reliability and completeness of sensor data can be easily compromised by various factors, such as sensor failures, communication errors, and operation in extreme environments. In particular, the complete loss of data from a failed sensor can disrupt the proper functioning of a predictive maintenance system that relies on multiple sensors. This indicates that even the loss of a single sensor’s data can result in incomplete datasets, which may significantly degrade the performance of AI models [5]. Deep learning models, which improve prediction accuracy by learning patterns from large volumes of high-quality training data, are especially vulnerable to such issues. When certain data channels are entirely missing or contain errors, the model’s predictive accuracy can drop drastically. This not only negatively impacts the reliability and safety of the equipment but also increases the risk of incorrect predictions, which may lead to abnormal operations or severe accidents. Therefore, developing robust AI models capable of handling sensor failures is critical.

A robust AI system must provide reliable predictions even in the presence of sensor failures, and research has been conducted to utilize alternative data sources or predictive techniques to achieve this goal [6,7].

To address these challenges, this study proposes a novel approach by applying topological data analysis (TDA) to develop an AI model that is robust against sensor failures. TDA is a technique that analyzes the structural properties of data to enhance robustness to missing data. It is expected to contribute to delivering reliable predictions in multi-sensor predictive maintenance systems [8,9,10].

## 2. Related Work

### 2.1. Multivariate Time Series Imputation

In real-world data, missing data are a frequent issue that can reduce the accuracy of data analysis. To address this, various methods have been proposed, including interpolation and traditional imputation techniques. Recently, deep-learning-based approaches for handling missing values have also been studied extensively [11,12,13,14,15,16]. For instance, Cao et al. (2018) proposed a deep learning model for imputing missing values in time series data which significantly improved classification performance [17].

Among deep-learning-based imputation methods, the use of Autoencoder-based models such as VAEI and its variant, the MIWAE, is noteworthy. VAEI (Variational Autoencoder Imputation) employs a Variational Autoencoder (VAE) to replace missing values by learning the latent space of the data and estimating the missing values [18].

VAEI offers advantages in generalization across diverse scenarios by learning the probabilistic distribution of the data, producing natural and consistent imputations for missing data. However, due to the probabilistic nature of the latent space in the VAE, the uncertainty in imputing missing values can be significant. As a result, the imputed values may lack consistency, and discrepancies from the original data distribution may arise if training is insufficient.

The MIWAE (Missing Data Importance-Weighted Autoencoder) is a VAE-based model designed to generate imputed values by assigning importance weights to missing data. Through importance-weighted sampling, the MIWAE aims to produce more accurate imputations [19]. This model operates robustly in scenarios with high missing rates or imbalanced data, maintaining key patterns in the data while imputing missing values. However, as the missing rate increases, the performance of the importance-weighted imputation may degrade, and there is a risk of overfitting to data points with higher weights, which could hinder generalization.

### 2.2. Multivariate Time Series Classification

Multivariate time series classification (MTSC) involves analyzing multiple time series variables simultaneously to learn patterns in the data and assign them to specific classes. MTSC is applied in various domains, such as finance, healthcare, and industrial monitoring [20,21,22,23]. Compared to univariate time series analysis, MTSC requires consideration of interactions between variables. For example, Bagnall et al. (2017) compared various approaches for learning multivariate features and proposed preprocessing techniques for precise pattern detection [24]. Recent studies have shifted from traditional statistical methods to deep learning models to extract more sophisticated features.

MTSC models can be broadly categorized into traditional machine learning algorithms and modern deep learning algorithms. Traditional methods, such as Random Forest and Support Vector Machines (SVM), have been widely used but face limitations in effectively handling multivariate characteristics [25,26]. On the other hand, deep learning models leverage multi-layer structures to efficiently learn data representations. Wang et al. (2017) proposed a Residual Network (ResNet) architecture, which has been evaluated as particularly effective for multivariate time series data as it extracts deep features through multiple layers [27]. Additionally, LSTM (Long Short-Term Memory) and CNN (Convolutional Neural Network)-based models have also been actively explored for multivariate time series analysis [28,29].

Recent research has focused on developing methods to efficiently handle multivariate time series data using deep learning models. Karim et al. (2019) proposed a hybrid architecture combining a CNN and LSTM to learn multi-channel time series data [30]. This architecture effectively captures inter-channel dependencies and has demonstrated superior performance across various applications. Furthermore, Transformer-based models have recently been introduced to time series analysis, opening new possibilities for solving MTSC problems [31,32,33].

### 2.3. Topological Data Analysis

Topological data analysis (TDA) is a powerful analytical method that applies topological concepts to analyze the morphological characteristics of data [34,35]. TDA is useful for deeply understanding the structure and patterns of data and uncovering hidden meanings, particularly capturing critical information that traditional statistical methods or machine learning algorithms might overlook. The core of TDA lies in understanding the topological “shape” of data, enabling the extraction of meaningful structural features from high-dimensional data.

The advantage of TDA lies in its ability to efficiently analyze complex structures in data [36,37,38]. It is particularly useful for high-dimensional or non-Euclidean data and exhibits robustness against noise. Recent studies applying TDA to multivariate time series classification (MTSC) aimed to extract the topological features of data, enabling clearer identification of complex patterns in multivariate time series data. For example, Gidea et al. (2018) utilized persistent homology to extract topological features from financial time series data and applied them for classification and prediction [39]. These studies demonstrate the potential of TDA in solving MTSC problems. With its robustness to noise making, it is well suited for scenarios involving sensor failures. This suggests that TDA can play a critical role in MTSC tasks, and its inherent robustness to noise is expected to provide more stable performance in the presence of sensor faults compared to conventional probabilistic imputation methods, which often suffer from high uncertainty and risk of overfitting.

## 3. Methodology

The proposed model architecture, depicted in Figure 1, includes two backbone models: a multivariate time series classifier for MTSC and a feature vector generation layer based on topological data analysis (TDA). These are followed by a feature vector classifier that analyzes the generated feature vectors. Both the MTSC Layer and the feature vector classifier employ basic models rather than advanced time series analysis techniques, to eliminate the influence of specific methodologies on the proposed approach.

In the MTSC Layer, the backbone model takes multivariate time series data as input, learns their temporal characteristics, and performs classification. The output probabilities for each class are computed through a Softmax layer. On the other hand, the TDA Layer considers the data at each time step *t* as an *m*-dimensional point and computes a persistent homology on the *m*-dimensional point cloud to generate feature vectors. These feature vectors reflect the topological connectivity of data points, providing additional information about the data’s structural form that is difficult to capture using traditional time series analysis. The generated feature vectors are fed into a classifier, which calculates class probabilities similarly to the MTSC Layer. Finally, the probabilities from both layers are combined using weighted summation, allowing the model to incorporate both temporal and topological features of the data.

### 3.1. Calculate Persistent Homology

#### 3.1.1. Simplex

A simplex is the most fundamental structure in TDA, used to describe the geometric relationships among data points in high-dimensional spaces. A simplex is a geometric object formed by connecting multiple data points, which can extend from being zero dimensional to higher dimensions. In TDA, simplices simplify complex data structures and help analyze their topological characteristics.

Simplices are defined by their dimensions, as illustrated in Figure 2, which shows simplices that range from zero dimensional to three dimensional. A zero-dimensional simplex is the most basic simplex, represented by a single point, which corresponds to individual data points in a dataset. A one-dimensional simplex is a line segment formed by connecting two points, a two-dimensional simplex is a triangle formed by connecting three points, and a three-dimensional simplex is a tetrahedron formed by connecting four points. This concept can be extended to higher dimensions, allowing the representation of complex data in higher-dimensional spaces.

#### 3.1.2. Simplicial Complex

A simplicial complex is a mathematical structure formed by combining multiple simplices to represent high-dimensional data and their interconnections. For a set of data points $V=\{v_{1},v_{2},\cdots ,v_{n}\}$ and a collection of subsets S, a simplicial complex is defined as (V,S), satisfying the following conditions:$$\begin{array}{c} (V,S)\;\mathrm{is}\;a\;\mathrm{simplicial}\;\mathrm{complex}\;\mathrm{if}\;\\ 1.\;\forall v\in V,\{v\}\in S\\ 2.\;\tau \subseteq \sigma \in S\Rightarrow \tau \in S \end{array}$$

These conditions indicate that every data point v∈V forms a zero-dimensional simplex within the simplicial complex and that any subset τ of a simplex σ in S also belongs to S.

To quantify the topological characteristics of simplices within a simplicial complex, Betti numbers are calculated. The Betti number $\beta_{k}$ represents the number of *k*-dimensional holes or voids. For instance, $\beta_{0}$ denotes the number of connected components, and $\beta_{1}$ denotes the number of holes, as illustrated in Figure 3.

#### 3.1.3. Filtration

Persistent homology tracks the changes in Betti numbers during the formation of a simplicial complex as data points are connected under a specific rule, a process known as filtration. One common filtration method is the Vietoris–Rips filtration, which constructs a simplicial complex by connecting data points within a distance 2ϵ, as shown in Figure 4.

During filtration, the simplicial complexes at each stage form a sequence of inclusions:$$\begin{array}{c} Ø=V_{0}\subseteq V_{1}\subseteq V_{2}\subseteq \cdots \subseteq V_{n}=V \end{array}$$

Here, $V_{i}$ represents the simplicial complex at stage *i*, where $V_{0}$ denotes the empty set or minimal connectivity, and $V_{n}$ represents the topological space where all data points are connected.

#### 3.1.4. Persistent Homology

Persistent homology tracks the Betti numbers of each simplicial complex throughout filtration, as illustrated in Figure 5.

As Betti numbers change, the birth and death of each dimension’s components are recorded as birth–death pairs. Persistent homology consists of these pairs for each dimension. A persistence diagram, as shown in Figure 6, visualizes the results.

The *x*-axis of the persistence diagram represents birth, while the *y*-axis represents death. The diagram in Figure 6 visualizes the birth and death of zero-dimensional points, one-dimensional edges, and two-dimensional faces.

### 3.2. Calculate Persistent Diagram Heatmap

In this study, we adopt a vectorization approach called the Persistence Diagram Heatmap to process the persistent homology computed in the TDA Layer. This method can be applied without considering dimensions explicitly; however, to observe the characteristics of each dimension more precisely, we constructed the layer by combining feature vectors computed for each dimension.

#### Persistence Diagram Heatmap

A Persistence Diagram Heatmap visualizes the birth–death pairs from the persistence diagram in a 2D space and applies a heatmap using a kernel. For each birth–death pair $\left(b_{i},d_{i}\right)$ from the persistence diagram, the heatmap kernel value at a grid coordinate (x,y) is computed as follows:(1)$$\begin{array}{c} H\left(x,y;b_{i},d_{i}\right)=\exp\left(-\frac{{\left(x-b_{i}\right)}^{2}+{\left(y-d_{i}\right)}^{2}}{2\sigma^{2}}\right) \end{array}$$

Here, σ denotes the standard deviation of the Gaussian kernel. The value decreases exponentially as the distance between the grid coordinate (x,y) and the point $\left(b_{i},d_{i}\right)$ increases, and increases as the distance decreases. To construct the heatmap, a grid of size r×r within the range [0,1]×[0,1] is defined, with grid coordinates $\left(x_{j},y_{l}\right)$ calculated as follows:$$\begin{array}{c} x_{j}=\frac{j}{r},\;y_{l}=\frac{l}{r},\;\mathrm{for}\;j,l=0,1,2,\ldots ,r-1 \end{array}$$

The resulting Persistence Diagram Heatmap $H(x_{j},y_{l})$ is computed as follows:(2)$$\begin{array}{c} H\left(x_{j},y_{l}\right)=\sum_{i=1}^{n}\exp\left(-\frac{{\left(x_{j}-b_{i}\right)}^{2}+{\left(y_{l}-d_{i}\right)}^{2}}{2\sigma^{2}}\right) \end{array}$$

Here, *n* is the number of birth–death pairs in the persistence diagram. This computation is applied to all dimensions, allowing the visualization of topological structures for each dimension. For example, a zero-dimensional heatmap reflects the birth and death of data points, a one-dimensional heatmap captures the connectivity and loops among data points, and a two-dimensional heatmap visualizes the formation and destruction of surfaces. The heatmap for the *k*-th dimension $H_{k}\left(x,y\right)$ is computed as follows:(3)$$\begin{array}{c} H_{k}\left(x,y\right)=\sum_{i=1}^{n_{k}}\exp\left(-\frac{{\left(x-b_{i,k}\right)}^{2}+{\left(y-d_{i,k}\right)}^{2}}{2\sigma^{2}}\right) \end{array}$$

Here, $n_{k}$ is the number of points in the *k*-th dimension’s persistence diagram, and $b_{i,k}$ and $d_{i,k}$ are the birth and death coordinates of the *k*-th dimension’s data points. Finally, the Persistence Diagram Heatmaps for all dimensions are combined into a *K*-channel tensor:$$\begin{array}{c} H=\{H_{0}\left(x,y\right),H_{1}\left(x,y\right),H_{2}\left(x,y\right),\cdots ,H_{K}\left(x,y\right)\} \end{array}$$

H represents the Persistence Diagram Heatmaps as a single *K*-channel image, as shown in Figure 7, enabling the topological features of multivariate time series data to be learned through an image classifier.

The Persistence Diagram Heatmap intuitively represents the lifespan of topological features, making it easier to identify key structural characteristics of data without significant information loss. Additionally, by tuning parameters, it is possible to flexibly emphasize either local or global features of the data.

### 3.3. Weighted Sum of MTSC Layer and TDA Layer

The proposed model integrates the feature vector output from the TDA Layer with the output of the MTSC backbone model to achieve enhanced classification performance. In this process, the probabilities generated by the MTSC Layer and the TDA Layer are linearly combined according to a weighting factor *w*, as expressed in Equation (Equation 4):(4)$$\begin{array}{c} P=w\cdot P_{MTSC}+\left(1-w\right)\cdot P_{TDA} \end{array}$$

Here, the probabilities from each layer are assigned weights, and the weighted sum is normalized to ensure the final result is a probability distribution summing to 1. This linear combination method leverages the complementary strengths of the two layers. The MTSC Layer excels at learning the overall patterns in the time series data, while the TDA Layer effectively captures the topological structures in the data, which reflect abnormal variations or fluctuations that may occur in the time series. By combining the strengths of different models based on the data characteristics and problem at hand, this approach provides more stable and improved performance compared to using a single model.

The proposed weighted summation method optimizes the complementary roles of MTSC and TDA, enabling robust classification performance across various data environments. Although mathematical optimization techniques could have been employed to identify the optimal combination ratio, this study primarily aims to explore the applicability of TDA in addressing the missing data problem in MTSC. Therefore, a heuristic experimental approach is adopted to analyze the effect of varying layer weight ratios.

## 4. Experiment

### 4.1. Experiment Setting

#### 4.1.1. Evaluation Method

To evaluate the performance of the proposed model for classifying multivariate time series data, standard performance metrics such as Accuracy, Precision, Recall, and F1 Score were employed. These metrics played a crucial role in comprehensively measuring the model’s classification capability from various perspectives, providing a reliable basis for assessing its performance under different scenarios.

A key focus of this study is to analyze how the model performance changes when information from certain channels is randomly removed. For this purpose, one or two channels were randomly excluded, and the model’s ability to maintain classification performance despite the missing information was evaluated. The performance under these scenarios was compared against the baseline scenario where all channels are utilized. This comparison provided insights into the model’s robustness and resilience against partial data loss. All experiments were conducted under the assumption that the population follows a normal distribution, with statistical tests performed at a 5% significance level.

#### 4.1.2. Dataset

The dataset used in this study was the “Rotating Machinery Type AI Dataset”, provided by the Ministry of SMEs and Startups, Korea AI Manufacturing Platform (KAMP), and the Department of Mechanical Engineering, KAIST (Daejeon, Republic of Korea). The dataset was accessed on 27 December 2021 via the official platform: https://www.kamp-ai.kr.

The data were acquired using the ERA (Educational Rotor Application) Test Station, a registered product of Signalink Co. (Seoul, Republic of Korea), shown in Figure 8 at a rotational speed of 1500 RPM. The detailed specifications of the rotor testbed are presented in Table 1.

The dataset was categorized into four states: Normal (type 1), Unbalance (type 2), Mechanical Looseness (type 3), and Unbalance with Mechanical Looseness (type 4). Unbalance refers to a state where the rotor’s mass center does not align with the center of rotation (shaft center), a significant cause of vibration in rotating machinery. Mechanical Looseness occurs when the machinery is not securely fixed to the ground or platform, resulting in horizontal misalignment.

Samples of the training data for each state are shown in Figure 9.

The raw dataset comprises vibration signals collected over 140 seconds from four different parts of the rotor testbed (Figure 8) using smart vibration sensors with the specifications listed in Table 2.

Each sensor was attached to a different section of the testbed, and the collected data included a time axis and four vibration channels, resulting in a total of five columns. This configuration produced a dataset with 754,477 rows, amounting to 3,772,385 data points in total. For model training, the data were segmented into windows of size 100 with an equal stride length. The resulting sliding window dataset was then split into training, validation, and test sets in a ratio of 80%, 10%, and 10%, respectively, to ensure reliable performance evaluation.

#### 4.1.3. Configuration

In this study, a simple 1D CNN was employed as the backbone for the MTSC Layer, while ResNet-152 was used as the backbone for the image classifier. These backbone models were combined with the TDA Layer to evaluate the performance of the proposed model under different weighted summation ratios and assess its robustness in scenarios with missing sensor data. The objective was to systematically analyze how the complementary strengths of MTSC and TDA influence the model’s performance and how the model behaves under varying layer weight ratios.

The 1D CNN model used for the MTSC Layer was designed with computational efficiency in mind, consisting of four consecutive convolutional blocks. Each block includes a 1D convolutional layer with a kernel size of 2 and stride of 1, followed by batch normalization, ReLU activation, max pooling with a kernel size of 1, and dropout for regularization. The first three blocks use 100 filters, and the final block projects to the number of output classes. An adaptive average pooling layer reduces the temporal dimension, followed by a fully connected linear layer for classification. The full architecture is summarized in Table 3.

For image classification based on topological signatures, the standard ResNet-152 architecture provided by PyTorch (version 2.1.2) was adopted. ResNet-152 is a deep convolutional neural network composed of an initial 7 × 7 convolutional layer, followed by four main stages of bottleneck blocks with identity and projection shortcuts. The model comprises a total of 152 layers, enabling efficient training of very deep architectures via residual learning. A global average pooling layer and a fully connected output layer are used at the final stage to produce class probabilities. An overview of the architecture is provided in Table 4.

The first experiment focused on evaluating performance based on weighted summation ratios. Five different ratios were tested, ranging from using only the MTSC Layer to using only the TDA Layer. This allowed for a quantitative assessment of how the layer weighting impacts model robustness.

Next, the impact of parameter variations in the TDA vectorization method on performance was analyzed. Comparisons were made between scenarios where all channels were used and where one channel was randomly removed, to evaluate how parameter changes in the vectorization method affect performance and how well the model maintains performance under missing channel conditions.

Additionally, the proposed TDA model was compared against statistical imputation methods, such as KNN and Mean Imputation, as well as deep-learning-based imputation models, including VAEI and the MIWAE. Further experiments were conducted on a high-dimensional dataset using the BJTU-RAO Bogie Dataset, which includes nine-channel gearbox vibration data, to explore the scalability of the proposed TDA model to higher-dimensional problems [40].

All training and experiments were conducted under the conditions specified in Table 5. Performance testing under identical conditions (excluding the batch size) showed that processing the entire dataset took approximately 12 minutes. This translates to an average processing time of about 300 ms per single data sample, shaped as a tensor with a (1,4,100) shape. Recognizing that data decentralization is an increasingly important issue in real-world industrial environments, recent studies have explored its implications and solutions. For instance, Yang et al. (2024) proposed a targeted transfer learning framework using a distribution barycenter medium to address the challenges of decentralized data in intelligent fault diagnosis [41]. Inspired by such approaches, future extensions of the proposed method could benefit from decentralized processing strategies, which may potentially reduce computation time by enabling parallel or distributed model execution across local nodes.

### 4.2. Experiment Results

#### 4.2.1. Experimental Results According to Weighted Sum Ratio

The experimental results based on the weighted sum ratio using the Persistence Diagram Heatmap, which demonstrated better performance in previous experiments, are presented in Table 6, Table 7 and Table 8. These experiments measured the model’s performance from cases where all channels were utilized to cases where up to two channels were removed. The weighted sum ratio was adjusted across five levels starting with 1.0 (using only the MTSC Layer), then gradually increasing the weight of the TDA Layer to 0.7, 0.5, 0.3, and, finally, 0.0 (using only the TDA Layer). Parameters for computing the Persistence Diagram Heatmap were fixed at σ=1.0 and resolution r=128.

When all channels were utilized, the model achieved the highest classification performance with the MTSC Layer alone (weight ratio 1.0). However, as channels were removed, the performance degraded significantly, indicating that the MTSC Layer alone lacks robustness to channel loss.

In contrast, using only the TDA Layer (weight ratio 0.0) resulted in relatively lower classification performance but exhibited only a slight average performance drop of about 8–9% when channels were removed. This demonstrates that the TDA Layer provides better robustness under channel loss conditions, ensuring more stable classification performance.

The performance based on the weighted sum ratio showed that increasing the MTSC Layer’s weight led to higher basic classification performance. However, similar to the MTSC Layer alone, the performance dropped sharply (up to approximately 70%) when channels were removed. The weighted sum ratio of 0.3 provided the best balance between classification accuracy and robustness. While its classification performance was relatively low at 84% when all channels were used, it maintained a higher performance of approximately 74% and 62% when one or two channels were removed, respectively. This indicates that the weighted sum ratio of 0.3 achieves the highest robustness compared to other ratios.

Therefore, these experimental results highlight that, while the MTSC Layer excels in classification performance, it is vulnerable to channel loss. Conversely, the TDA Layer is robust to channel loss but has lower classification accuracy. Combining these layers ensures stable classification performance under channel loss conditions, enhancing the model’s overall reliability.

#### 4.2.2. Experimental Results According to Parameters

The experimental results evaluating the changes in key parameters—standard deviation σ and resolution *r*—for computing the Persistence Diagram Heatmap are presented in Table 9. These experiments were conducted based on the weighted sum ratio of 0.3, which demonstrated the highest robustness in previous experiments. The F1 Score was used as the evaluation metric.

The results show that the parameter combination σ=1.0 and r=512 achieved the highest performance in both scenarios: 92.41% when all channels were utilized, and 81.83% when a single channel was removed. However, this combination exhibited a performance drop of approximately 11% between the two scenarios. In contrast, the combination σ=0.1 and r=512 showed a smaller performance drop of around 6%, indicating better robustness. These findings suggest that higher resolutions (*r*) improve classification performance, while lower standard deviations (σ) increase robustness to channel loss.

Higher resolutions enhance classification performance by generating more detailed and sharper heatmaps, which effectively capture complex data patterns. On the other hand, lower standard deviations create localized heatmaps that are more resistant to changes in data structure, thereby maintaining stability under channel loss conditions. Conversely, as the Gaussian Kernel’s standard deviation increases, the heatmap captures features over a wider area, making it more sensitive to global changes in the dataset, which increases vulnerability to channel loss.

Thus, the choice of standard deviation σ and resolution *r* is a critical factor in balancing classification performance and robustness to channel loss. Proper tuning of these parameters ensures the design of a robust model capable of achieving high classification performance even under missing channel conditions.

### 4.3. Performance Comparison with Other Models

When a single channel was removed, the performance of the proposed TDA model was compared with existing imputation models using the F1 Score. For data imputation, KNN, Mean, VAEI, and MIWAE methods were applied, and the imputed data were fed into the MTSC Layer for classification tasks. The learning rate, batch size, and epochs for each method were set identically to those used in the TDA model. For the TDA model, performance was evaluated by combining a weight ratio of 0.3, standard deviation σ=0.5, and resolutions r=128, r=256, and r=512. The results are summarized in Table 10.

The experimental results show that the F1 Scores of data imputed using KNN and Mean methods were approximately 63% and 50%, respectively. In contrast, the VAEI and MIWAE methods achieved higher F1 Scores of around 71% and 76%, respectively, demonstrating better performance. These results are comparable to the TDA model’s performance with a resolution of r=128. However, as the resolution increased to r=256 and r=512, the TDA model outperformed both VAEI and the MIWAE.

This trend indicates that higher resolutions in the TDA model allow it to surpass the performance of VAEI and the MIWAE by capturing complex patterns in the data more effectively, thus improving classification performance.

Meanwhile, the lower F1 Scores observed with KNN and Mean methods can be attributed to their inability to capture the intricate data structures. These methods simply replace missing data without reflecting the underlying patterns, resulting in relatively poor performance. Although VAEI and the MIWAE leverage neural-network-based imputation to better capture the latent structure of the data, they fall short of the detailed feature extraction provided by the TDA model at higher resolutions.

### 4.4. Experiment with Additional Data

The results of additional experiments conducted using a subset of the BJTU-RAO Bogie Dataset, consisting of nine-channel gearbox vibration data, are presented in Table 11 and Table 12. These experiments were evaluated using the same metrics as the first experiment: Accuracy, Precision, Recall, and F1 Score. The weighted sum ratios were set to three levels (1.0, 0.5, and 0.0), and the performance was compared between cases where all channels were utilized and where three random channels were removed.

The experimental results indicate that, when all channels were utilized, the highest performance was achieved with a weighted sum ratio of 1.0, showing a similar trend to the first experiment. In contrast, when three channels were removed, the best performance was observed with a weighted sum ratio of 0.5, while the lowest performance degradation occurred with a weighted sum ratio of 0.0.

These results demonstrate that the proposed TDA model is scalable and robust for high-dimensional data. Specifically, the TDA model effectively learns complex patterns and adapts flexibly to missing data, as evidenced by the optimal performance achieved with a weighted sum ratio of 0.5 and the minimal performance degradation observed with a ratio of 0.0.

In this experiment, the nine-channel high-dimensional data yielded performance patterns similar to those observed in the four-channel data experiments. This further validates the proposed model’s suitability for processing high-dimensional data.

## 5. Conclusions

In conclusion, this paper proposes a robust AI model based on an MTSC model combined with TDA, designed to withstand sensor data loss. The experimental results showed that the model maintained the performance of traditional MTSC while experiencing only about a 10% performance drop in cases where approximately 25% of the sensors used in the data were missing. This demonstrates superior performance retention compared to using the MTSC model alone. In particular, the results confirmed the applicability of TDA in handling data deficiency problems, with observed performance improvements ranging from at least 4% to up to 30% over the baseline MTSC model depending on the missing sensor scenarios.

Future research will explore not only the heatmap kernel but also additional TDA methodologies, aiming to extend the potential of TDA beyond simple classification problems to a wider variety of tasks.

## Figures and Tables

**Figure 1 sensors-25-02709-f001:**
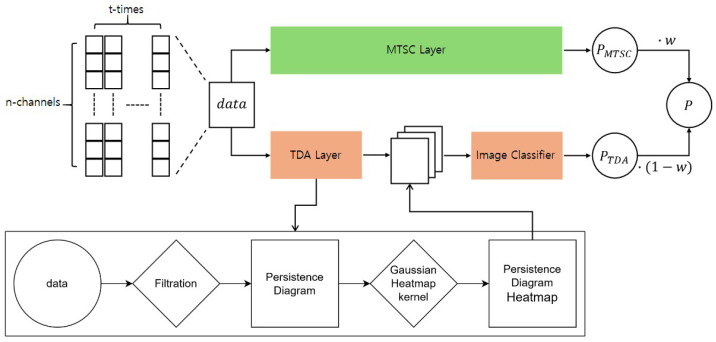
Architecture of the proposed model. The input data are processed in parallel through two separate layers, and the classification results from each layer are combined using a weighted fusion strategy.

**Figure 2 sensors-25-02709-f002:**
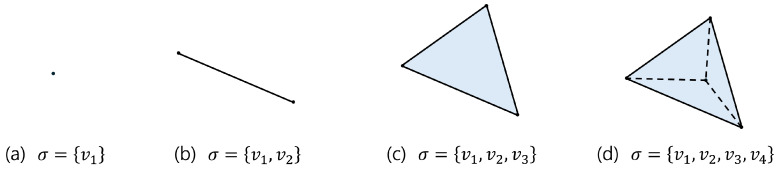
(**a**) 0 simplex, (**b**) 1 simplex, (**c**) 2 simplex, (**d**) 3 simplex.

**Figure 3 sensors-25-02709-f003:**
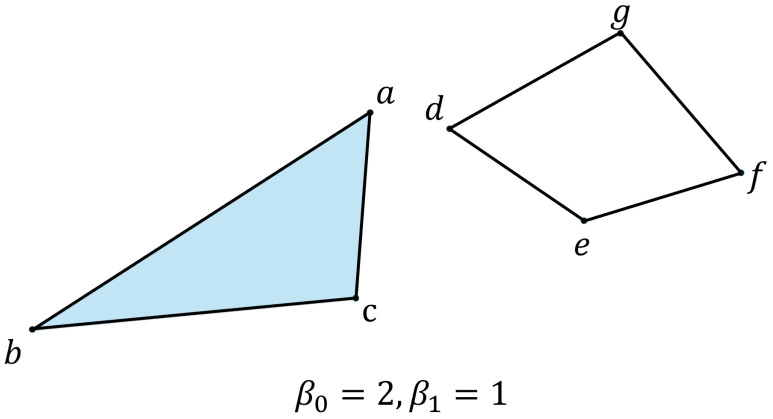
The set {a,b,c} is a connected component, as is {d,e,f,g}. The set {d,e,f,g} constitutes a hole.

**Figure 4 sensors-25-02709-f004:**
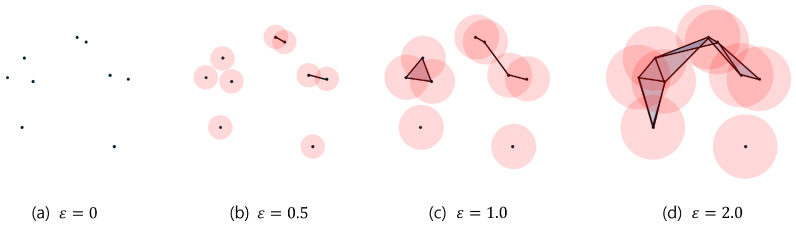
Data points are connected as distance ϵ increases: (**a**) no connections, (**b**) connections within distance 1.0, (**c**) connections within distance 2.0, (**d**) connections within distance 4.0.

**Figure 5 sensors-25-02709-f005:**
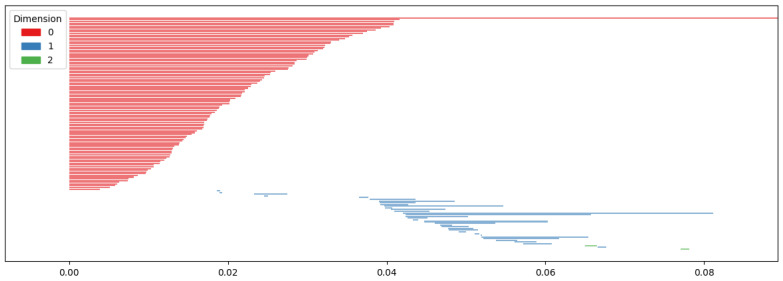
As ϵ changes following filtration, a connection is created between each data point, and the change in Betti number is tracked accordingly.

**Figure 6 sensors-25-02709-f006:**
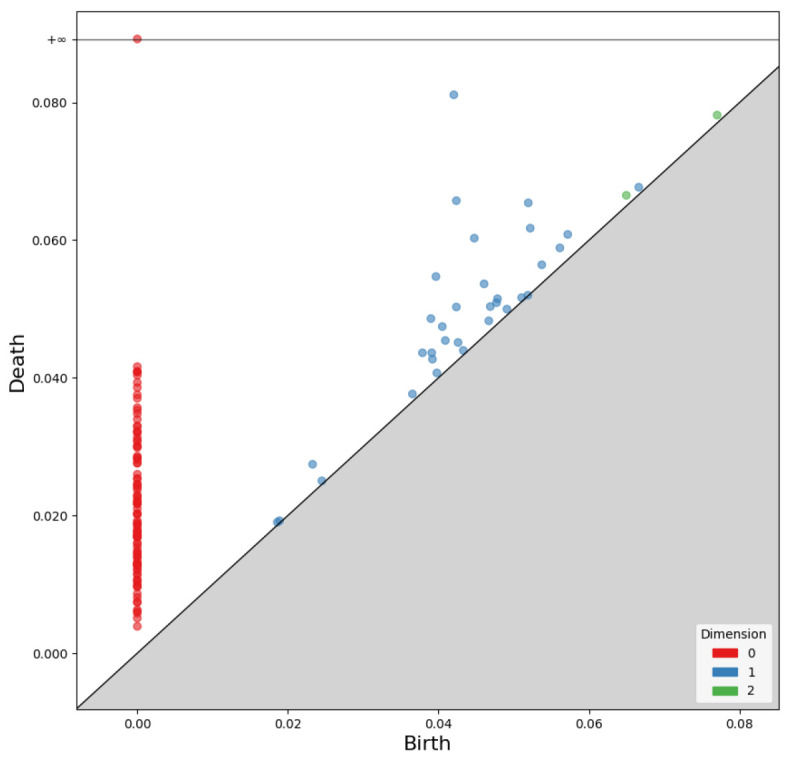
Two-dimensional visualization of persistence homology, where the *x*-axis represents birth and the *y*-axis represents death.

**Figure 7 sensors-25-02709-f007:**
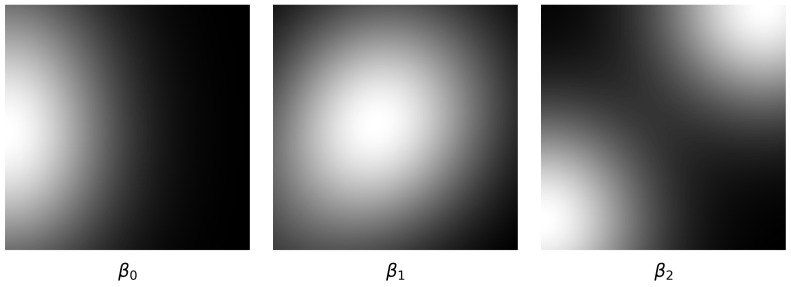
Visualization of Persistence Diagram Heatmap calculated based on Betti number.

**Figure 8 sensors-25-02709-f008:**
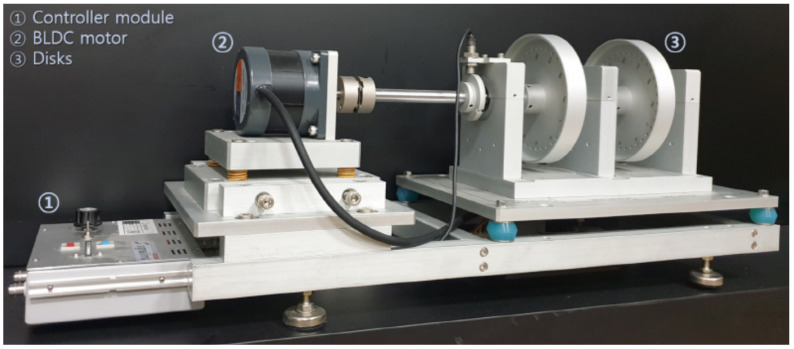
Rotor testbed.

**Figure 9 sensors-25-02709-f009:**
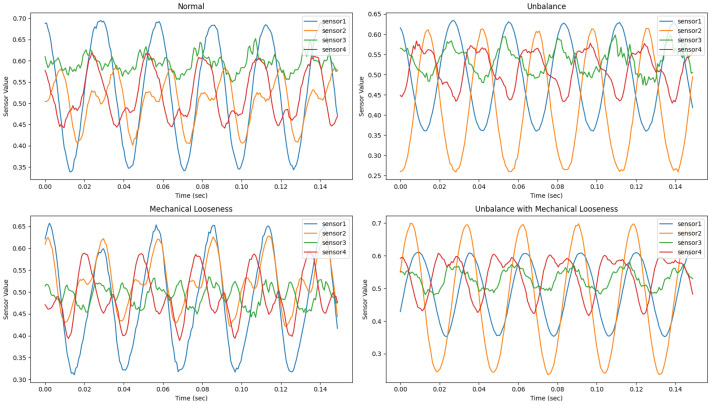
Training data samples.

**Table 1 sensors-25-02709-t001:** Rotor testbed specifications.

Parameter	Specification
Size	673 mm (W) × 280 mm (D) × 281 mm (H)
Weight	25 kg
Material	Aluminum
Bearing	6202ZZ × 2EA
Motor	DC 12 V × 0.25 HP (0.2 kW), 0~3000 RPM
Main power	220 VAC

**Table 2 sensors-25-02709-t002:** Specifications of the vibration sensor.

Parameter	Specification
Size	44 mm (W) × 44 mm (D) × 25 mm (H)
Weight	About 40 gf
Material	PC (Polycarbonate)
Fixing Method	M4 Bolt or Magnetic
Frequency Span	672 Hz
Frequency Resolution	1.3125 Hz

**Table 3 sensors-25-02709-t003:** Architecture of the 1D CNN used in the MTSC Layer.

Layer	Type	Output Channels	Kernel/Stride/Padding
Conv1	Conv1d	100	2/1/same
Conv2	Conv1d	100	2/1/same
Conv3	Conv1d	100	2/1/same
Conv4	Conv1d	5 (output)	2/1/same
Pooling	AdaptiveAvgPool1d	-	-
FC	Linear	5	-

**Table 4 sensors-25-02709-t004:** Overview of the ResNet-152 architecture used for image classification.

Stage	Layers (Bottleneck Blocks)	Output Size
Conv1	7 × 7 Conv, 64, stride 2 + MaxPool 3 × 3	112 × 112
Conv2_x	3 Bottleneck Blocks	56 × 56
Conv3_x	8 Bottleneck Blocks	28 × 28
Conv4_x	36 Bottleneck Blocks	14 × 14
Conv5_x	3 Bottleneck Blocks	7 × 7
Pooling	Global Average Pooling	1 × 1
FC	Fully Connected	Number of Classes

**Table 5 sensors-25-02709-t005:** Training and testing conditions.

Parameter	Value
Framework	PyTorch
GPU	1 NVIDIA GeForce RTX 2080 Ti
Training Batch Size	256
Training Data Shape	(256,4,100)
Training Epochs	1000
Learning Rate	0.01
Testing Batch Size	1
Testing Data Shape	(1,4,100)

**Table 6 sensors-25-02709-t006:** Performance when all channels were utilized.

Metric	Ratio 1.0	Ratio 0.7	Ratio 0.5	Ratio 0.3	Ratio 0.0
Accuracy	**98.81**	97.67	92.13	84.25	75.00
Precision	**98.90**	98.15	92.76	84.40	76.36
Recall	**98.63**	97.24	89.85	83.90	74.29
F1 Score	**98.76**	97.68	91.28	84.15	75.31

**Table 7 sensors-25-02709-t007:** Performance when a single channel was removed.

Metric	Ratio 1.0	Ratio 0.7	Ratio 0.5	Ratio 0.3	Ratio 0.0
Accuracy	26.75	36.63	50.50	**74.00**	66.25
Precision	27.36	38.54	54.44	**75.48**	67.87
Recall	26.20	34.85	49.24	**72.79**	64.10
F1 Score	26.77	36.60	51.71	**74.11**	65.93

**Table 8 sensors-25-02709-t008:** Performance when two channels were removed.

Metric	Ratio 1.0	Ratio 0.7	Ratio 0.5	Ratio 0.3	Ratio 0.0
Accuracy	24.50	26.25	40.38	**62.50**	58.88
Precision	24.63	28.35	44.98	**64.57**	60.28
Recall	22.27	22.97	38.88	**60.61**	55.37
F1 Score	23.39	25.38	41.71	**62.53**	57.72

**Table 9 sensors-25-02709-t009:** F1 Score for each parameter.

Parameter	Use All Channels	Remove Single Channel
σ=0.1,r=128	83.94	68.79
σ=0.1,r=256	86.64	79.71
** σ=0.1,r=512 **	**87.21**	**81.08**
σ=0.5,r=128	85.18	77.94
σ=0.5,r=256	87.23	79.17
σ=0.5,r=512	87.50	80.33
σ=1.0,r=128	84.15	74.11
σ=1.0,r=256	88.03	76.90
** σ=1.0,r=512 **	**92.41**	**81.83**

**Table 10 sensors-25-02709-t010:** F1 Score for each model.

Models	F1 Score
KNN	63.33
Mean	50.12
VAEI	71.81
MIWAE	76.04
TDA r=128	77.94
TDA r=256	79.17
**TDA r=512**	**80.33**

**Table 11 sensors-25-02709-t011:** All channels were utilized with the BJTU-RAO Bogie Dataset.

Metric	Ratio 1.0	Ratio 0.5	Ratio 0.0
Accuracy	**99.58**	98.77	64.16
Precision	**98.90**	98.15	62.88
Recall	**98.63**	97.24	62.07
F1 Score	**98.76**	97.68	62.47

**Table 12 sensors-25-02709-t012:** Three channels were removed with the BJTU-RAO Bogie Dataset.

Metric	Ratio 1.0	Ratio 0.5	Ratio 0.0
Accuracy	66.50	**80.47**	54.68
Precision	84.78	**88.48**	54.07
Recall	46.78	**74.11**	50.34
F1 Score	60.51	**80.66**	52.14

## Data Availability

The data presented in this study are available on KAMP (Korea AI Manufacturing Platform) at https://www.kamp-ai.kr, accessed on 27 December 2021.

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
