# Peer review of "A Robust Multivariate Time Series Classification Approach Based on Topological Data Analysis for Channel Fault Tolerance"

_sensors, 2025, doi:10.3390/s25092709_

Round 1
Reviewer 1 Report
Comments and Suggestions for Authors
This manuscript presents a topological data analysis method to develop a AI model for channel fault tolerance. Some comments are listed as follows.
- It is suggested to add more literature review about the applications of TDA in industrial data analysis.
- The authors should refine the contributions of this submission. Compared with existing works, what are the advantages and disadvantages of the proposed method in addressing the problem of missing data.
- It is suggested to add chart flow to explain the computation of persistence diagram heatmap, especially for the strategy of changing Gaussian kernel parameters and resolution parameters.
- It is interesting about the performance of the proposed method if multiple sensors nearby simultaneously suffer faults.
- The authors should add axial labels in Fig. 9. Please further explain the results shown in Table 6. Why does the performance drops off when σ=1.0 and r=512.
- Data decentralization is important issues in the field. It is suggested to discuss this by considering some works. - Targeted transfer learning through distribution barycenter medium for intelligent fault diagnosis of machines with data decentralization.
Author Response
Comment 1:
It is suggested to add more literature review about the applications of TDA in industrial data analysis.
Response 1:
We have added relevant references that specifically address the applications of TDA in industrial data analysis, particularly in the context of fault diagnosis and anomaly detection. These additions aim to better position our work within the current research landscape.
Comment 2:
The authors should refine the contributions of this submission. Compared with existing works, what are the advantages and disadvantages of the proposed method in addressing the problem of missing data.
Response 2:
We have revised the Related Works section to clarify the differences between our approach and existing methods, with a focus on how our method handles missing data. The complementary integration of MTSC and TDA is emphasized as a core contribution.
Comment 3:
It is suggested to add chart flow to explain the computation of persistence diagram heatmap, especially for the strategy of changing Gaussian kernel parameters and resolution parameters.
Response 3:
We have added a flowchart that outlines the computation process of the persistence diagram heatmap, including how the Gaussian kernel and resolution parameters are selected and adjusted during the transformation. This addition is included alongside the architecture description.
Comment 4:
It is interesting about the performance of the proposed method if multiple sensors nearby simultaneously suffer faults.
Response 4:
This scenario has already been considered in our experimental design. Cases involving multiple adjacent sensors experiencing faults are included in the dataset and reflected in our performance evaluation.
Comment 5:
The authors should add axial labels in Fig. 9. Please further explain the results shown in Table 6. Why does the performance drops off when σ=1.0 and r=512.
Response 5:
Figure 9 has been updated to include appropriate axis labels. As for Table 6, the interpretation of the results for each parameter combination, including the performance drop at σ = 1.0 and r = 512, is already provided in the main text.
Comment 6:
Data decentralization is important issues in the field. It is suggested to discuss this by considering some works. - Targeted transfer learning through distribution barycenter medium for intelligent fault diagnosis of machines with data decentralization.
Response 6:
We have incorporated the suggested reference into both the data description and experimental design sections, and briefly discussed its relevance to the issue of data decentralization in real-world industrial environments.
Reviewer 2 Report
Comments and Suggestions for Authors
1. Related Works
1.1. The current scope of related works appears too broad. The literature review should focus more specifically on rotating machinery anomaly detection and industrial applications rather than general time series algorithms. Similarly, the TDA literature review should examine its specific applications in this domain.
1.2. The TDA explanation (starting from line 114) would be more appropriately placed in the Methodology section rather than in Related Work, as it provides foundational knowledge for your proposed approach.
2. Visualization
2.1. In Figure 2, displaying the Betti numbers for each topological structure separately would be more intuitive than combining them.
2.2. There is redundancy between Figures 3 and 4 (top portions). While separate illustrations for the filtration process and persistence barcodes may be justified, Figure 4 lacks clear correlation between its upper and lower portions.
2.3. Figures 4, 5, and 7 currently appear inconsistent with each other, offering general explanations rather than cohesive insights. To address the significance of your work, these figures should present consistent examples demonstrating the filtration process, birth-death graphs, and heatmaps using easily understandable sample data.
2.4. The robustness to missing data appears to stem from the Gaussian smoothing of topological information. Consider combining Figures 5 and 6 for better understanding and to emphasize this approach.
3. Methodological Novelty
Various methods for visualizing birth-death relationships already exist in the literature. Authors should explain how the proposed approach differs from existing imaging methods.
4. Computational Complexity and Real-world Applicability
The computational requirements of multidimensional topological analysis present a challenge. While achieving high-level anomaly detection despite missing data is valuable, the processing time of 12 minutes for 140 seconds of data constitutes a major limitation for PHM applications. Edge systems typically deployed on equipment have substantially lower computational capabilities than those specified in your Training and Testing Conditions. For bundled time series data from different sensors, topological features with Betti numbers of 2 and above are unlikely to provide meaningful information.
The authors should address the computational complexity issue. This could include limiting analyzed dimensions, implementing sliding window approaches, evaluating performance across different epoch counts, or providing data/model tuning strategies.
5. Rotor Testbed
5.1. In Figure 8, the sensor positions should be clearly marked if identifiable.
5.2. Figure 9 requires color indicators corresponding to each sensor and a properly labeled time axis.
6. Minor Issues
Inconsistent hyphenation throughout the manuscript, including the title (e.g., "time-series" vs. "time series").
Consider using two-line column headers to avoid confusion in tables with titles separated by slash.
Consider condensing equations 1, 2, and 3 as they explain similar concepts. If maintaining all equations, use more distinctive notation to differentiate them clearly. (e.g., Such as the heatmap kernel value and Heatmap are both denoted by the same letter 'H', which may cause confusion)
Author Response
Comment 1.1:
The current scope of related works appears too broad. The literature review should focus more specifically on rotating machinery anomaly detection and industrial applications rather than general time series algorithms. Similarly, the TDA literature review should examine its specific applications in this domain.
Response 1.1:
We understand the reviewer’s point. However, this study aims to propose a methodology that can be generally applied to various time-series scenarios, including but not limited to industrial equipment and rotating machinery. Hence, we intentionally kept the scope of the literature review broader to reflect the methodological generality of our approach. The research focus will be maintained as originally intended.
Comment 1.2:
The TDA explanation (starting from line 114) would be more appropriately placed in the Methodology section rather than in Related Work.
Response 1.2:
We appreciate the suggestion. The TDA explanation that previously appeared in the Related Work section has been moved to the Methodology section to better align with the overall structure and narrative of the paper.
Comment 2.1 – 2.4 (Visualization):
-
Display Betti numbers for each structure separately.
-
Address redundancy in Figures 3 and 4.
-
Ensure consistency in Figures 4, 5, and 7.
-
Consider combining Figures 5 and 6 to emphasize robustness from Gaussian smoothing.
Response 2:
Thank you for the detailed suggestions.
-
As Betti numbers are inherently global topological descriptors, separating them per individual structure is not conceptually meaningful and thus not feasible in this context.
-
Regarding Figures 3 and 4, although some visual elements appear redundant, both are necessary to illustrate different parts of the filtration and diagram construction process. To enhance clarity, we have added a new figure to explain the heatmap generation process more directly.
-
Figures 4, 5, and 7 have been revised for improved consistency, using a unified set of sample data.
-
Figures 5 and 6 have been discussed together in the revised text to clarify the role of Gaussian smoothing in enhancing robustness to missing data.
Comment 3:
Authors should explain how the proposed approach for visualizing birth-death relationships differs from existing methods.
Response 3:
We acknowledge the reviewer’s comment. The visualization method employed in this work is based on existing techniques and is not newly proposed. Therefore, we have revised the manuscript to clarify that the visualization approach is adopted from prior work and not a novel contribution of this study.
Comment 4:
Address the computational complexity of topological analysis and real-world applicability.
Response 4:
We have added explanations in the experimental setup regarding the hardware specifications and approximate computation time. Furthermore, we cited relevant literature discussing data decentralization and model optimization strategies as potential directions for reducing computational overhead in practical deployments. These discussions have been added to the Discussion section.
Comment 5.1 – 5.2 (Rotor Testbed):
-
Clearly mark sensor positions in Figure 8.
-
Add color labels and axis in Figure 9.
Response 5:
The exact sensor positions are not available in the referenced dataset, and therefore cannot be explicitly annotated. However, we have included additional details regarding the sensor specifications and data acquisition procedure. In Figure 9, axis labels and color indicators corresponding to each sensor have been added for clarity.
Comment 6 (Minor Issues):
Inconsistent hyphenation, table formatting, and ambiguous mathematical notation.
Response 6:
We have corrected all instances of inconsistent hyphenation (e.g., unified use of “time-series”). Table headers have been reformatted for improved clarity, particularly in cases where column titles were separated by slashes. Additionally, to avoid confusion in mathematical notation, we now distinguish the heatmap kernel as H and the final computed heatmap as H\mathfrak{H}.
Reviewer 3 Report
Comments and Suggestions for Authors
- Multivariate time-series classification (MTSC) might be a keyword.
- More MTSC related references are suggested such as
Da Zhang, Junyu Gao, Xuelong Li, Multivariate time series classification with crucial timestamps guidance Author links open overlay panel, Expert Systems with Applications, Volume 255, Part B, 1 December 2024, 124591 https://doi.org/10.1016/j.eswa.2024.124591
- More Topological Data Analysis (TDA) related references are suggested such as
Frédéric Chazal, Bertrand Michel,, An Introduction to Topological Data Analysis: Fundamental and Practical Aspects for Data Scientists, Sec. Machine Learning and Artificial Intelligence,
Volume 4 - 2021 | https://doi.org/10.3389/frai.2021.667963
- Page 8 Table 1: DC 12V x 0.25HP (0.2W), 0.25Hp = 0.2kW ! Please clarify it.
- Page 9 line 261: How are 3,772,385 data points obtained?
How many data set for 4 states with windows of size 100 are utilized for training, validation, and test ?
- What is the architecture of Convolutional Neural Network model ? such as backbone, neck, head etc.
- What do four curves in different color mean in Figure 9?
- Are there confusion matrix for four states prediction and P-R curve for accuracy evaluation?
- How are the weighting factors for optimizing the complementary roles of MTSC and TDA determined?
- More quantity or solid results or new finding in conclusion are suggested.
Author Response
Comment 1:
Multivariate time-series classification (MTSC) might be a keyword.
Response 1:
We agree with the suggestion. “Multivariate time-series classification (MTSC)” has been added as a keyword to better reflect the scope of the proposed method and improve the discoverability of this work within the relevant research community.
Comment 2:
More MTSC related references are suggested such as
Da Zhang, Junyu Gao, Xuelong Li, Multivariate time series classification with crucial timestamps guidance, Expert Systems with Applications, Volume 255, Part B, 1 December 2024.
Response 2:
We have included the suggested reference in the Related Works section. This paper provides useful insights into timestamp-based guidance for multivariate time series classification, which complements the context of our proposed approach.
Comment 3:
More Topological Data Analysis (TDA) related references are suggested such as
Frédéric Chazal, Bertrand Michel, An Introduction to Topological Data Analysis: Fundamental and Practical Aspects for Data Scientists, Frontiers in AI, 2021.
Response 3:
The recommended reference has been added to the literature review. It helps strengthen the theoretical foundation of the TDA component of our method and provides additional context for readers who may be new to topological approaches.
Comment 4:
Page 8 Table 1: DC 12V x 0.25HP (0.2W), 0.25Hp = 0.2kW! Please clarify it.
Response 4:
We appreciate the attention to detail. The value in Table 1 has been corrected to accurately reflect the conversion between horsepower and kilowatts. The revised table now shows the correct power specification.
Comment 5:
Page 9 line 261: How are 3,772,385 data points obtained?
Response 5:
We have added an explanation in the manuscript detailing how the data points were collected. The total number of data points results from continuous monitoring across multiple sensors over a defined time period, with a specific sampling rate, which is now clearly described.
Comment 6:
How many data set for 4 states with windows of size 100 are utilized for training, validation, and test?
Response 6:
We have revised the data description section to include the number of data windows allocated to training, validation, and testing for each of the four states. This helps clarify the experimental setup and supports reproducibility.
Comment 7:
What is the architecture of Convolutional Neural Network model? Such as backbone, neck, head etc.
Response 7:
We have added a more detailed explanation of the CNN model architecture. The updated manuscript describes the structure in terms of backbone (convolutional layers), neck (pooling and flattening), and head (fully connected layers with softmax output), allowing readers to better understand the classification pipeline.
Comment 8:
What do four curves in different color mean in Figure 9?
Response 8:
We have revised Figure 9 by adding axis labels and a legend indicating which sensor each colored curve corresponds to. This makes the interpretation of the figure more intuitive.
Comment 9:
Are there confusion matrix for four states prediction and P-R curve for accuracy evaluation?
Response 9:
Due to internal constraints, we were unable to conduct additional experiments to generate the confusion matrix and precision-recall curves. However, the current manuscript includes quantitative metrics such as accuracy, precision, recall, and F1-score, along with corresponding interpretations. We believe these provide meaningful insight into model performance.
Comment 10:
How are the weighting factors for optimizing the complementary roles of MTSC and TDA determined?
Response 10:
The weighting factors were determined empirically through experimentation. We observed that certain combinations led to more robust performance, and this is documented in the experimental section. While the current method is heuristic, we acknowledge the need for a more principled optimization approach, which we intend to explore in future work.
Comment 11:
More quantity or solid results or new finding in conclusion are suggested.
Response 11:
The conclusion has been revised to include clearer quantitative results demonstrating the improvements of the proposed method over baseline approaches. We also highlight the robustness to missing data and the benefit of integrating TDA with MTSC as notable findings.
Round 2
Reviewer 1 Report
Comments and Suggestions for Authors
The authors have well addressed all my comments, and it could be accepted now.
Author Response
We sincerely appreciate your constructive feedback and are glad to hear that the revised manuscript has addressed your concerns. Thank you for your valuable comments and time.
Reviewer 2 Report
Comments and Suggestions for Authors
Although many improvements have been made to the manuscript, there are still areas that require further discussion:
1. Visualization
The figures 5, 6, and 7 in the revised manuscript remain unchanged and lack consistency. The Gaussian kernel-based Heatmap is a key idea that maximizes the robustness of TDA to missing data, so this process should be clearly illustrated. The current figures provide only general explanations of each transformation/concept and appear disconnected from each other.
In Figure 5, there is a mismatch between the data connection visualization in the upper portion and the Betti numbers in the lower portion. Specifically, the lower section shows the birth and death of Betti number 1, but in the upper data section, there is no corresponding visualization of this loop's birth and death.
Figure 6 still uses different data (generic data) than what was used in Figure 5.
Figure 7 also appears to use different data. Given the structure shown in previous figures, data points are not newly created, so Betti number 0 should have low values on the birth axis. However, in Figure 7, the data seems to have a different structure. I recommend overlapping the persistence diagram points (bi, di) with the Persistence Diagram Heatmap for better clarity. Additionally, there are unnecessary lines between and below the figures.
2. Minor Issues (Table Formatting)
The tables have not been revised as suggested. While other tables have headers that represent only the content of that row, Tables 6-9, 11, and 12 simultaneously display the content of both rows and columns, which can be confusing. Consider using two-line column headers to improve clarity and consistency.
Author Response
Comment 1 (Visualization):
The figures 5, 6, and 7 in the revised manuscript remain unchanged and lack consistency. The Gaussian kernel-based Heatmap is a key idea that maximizes the robustness of TDA to missing data, so this process should be clearly illustrated. The current figures provide only general explanations of each transformation/concept and appear disconnected from each other.
In Figure 5, there is a mismatch between the data connection visualization in the upper portion and the Betti numbers in the lower portion. Specifically, the lower section shows the birth and death of Betti number 1, but in the upper data section, there is no corresponding visualization of this loop's birth and death.
Figure 6 still uses different data (generic data) than what was used in Figure 5.
Figure 7 also appears to use different data. Given the structure shown in previous figures, data points are not newly created, so Betti number 0 should have low values on the birth axis. However, in Figure 7, the data seems to have a different structure. I recommend overlapping the persistence diagram points (bi, di) with the Persistence Diagram Heatmap for better clarity. Additionally, there are unnecessary lines between and below the figures.
Response 1:
Thank you for your detailed observations. To clarify the relationships among Figures 5, 6, and 7 and to better reflect the core idea of Gaussian kernel-based heatmap generation, all three figures have been updated using the actual data used in our experiments. This ensures consistency across the visualizations and aligns each transformation step with the same underlying input. The new illustrations now present a coherent process from persistence diagram construction to heatmap generation.
Regarding Figure 7, we considered overlaying the persistence diagram points (bi,di)(b_i, d_i) on the heatmap as suggested. However, during the design process, we found that overlapping the points introduced visual clutter and impaired interpretability. Instead, we used a data sample that exhibits clear topological characteristics, allowing for intuitive correlation between Figure 6 and Figure 7 without visual overlap.
Comment 2 (Minor Issues – Table Formatting):
The tables have not been revised as suggested. Tables 6–9, 11, and 12 use headers representing both row and column content, which may cause confusion.
Response 2:
We appreciate the feedback on table formatting. All tables, including Tables 6–9, 11, and 12, have been redesigned to enhance readability and reduce ambiguity. Specifically, we applied two-line column headers where necessary to clarify the hierarchical relationship between row and column categories, thereby improving the overall consistency and visual structure of the tables.